# A plausible identifiable model of the canonical NF-κB signaling pathway

**Joanna Jaruszewicz-Błońska**, **Ilona Kosiuk, Wiktor Prus, Tomasz Lipniacki** *

Institute of Fundamental Technological Research of the Polish Academy of Sciences, Warsaw, Poland

* tlipnia@ippt.pan.pl

## Abstract

An overwhelming majority of mathematical models of regulatory pathways, including the intensively studied NF-κB pathway, remains non-identifiable, meaning that their parameters may not be determined by existing data. The existing NF-κB models that are capable of reproducing experimental data contain non-identifiable parameters, whereas simplified models with a smaller number of parameters exhibit dynamics that differs from that observed in experiments. Here, we reduced an existing model of the canonical NF-κB pathway by decreasing the number of equations from 15 to 6. The reduced model retains two negative feedback loops mediated by IκBα and A20, and in response to both tonic and pulsatile TNF stimulation exhibits dynamics that closely follow that of the original model. We carried out the sensitivity-based linear analysis and Monte Carlo-based analysis to demonstrate that the resulting model is both structurally and practically identifiable given measurements of 5 model variables from a simple TNF stimulation protocol. The reduced model is capable of reproducing different types of responses that are characteristic to regulatory motifs controlled by negative feedback loops: nearly-perfect adaptation as well as damped and sustained oscillations. It can serve as a building block of more comprehensive models of the immune response and cancer, where NF-κB plays a decisive role. Our approach, although may not be automatically generalized, suggests that models of other regulatory pathways can be transformed to identifiable, while retaining their dynamical features.

**Data Availability Statement:** The computer codes of the original and reduced computational models developed in this study as well as of earlier NF-κB models are available as S1 Code.

## Introduction

The key and frequent drawback of mechanistic models of regulatory pathways is a lack of identifiability, meaning that model parameters may not be determined. One can distinguish between structural non-identifiability, when parameters are inherently indeterminable, and practical non-identifiability, arising when based on available data parameters may not be determined with adequate precision. The structural non-identifiability can be resolved by reparametrization and non-dimensionalization; however, the practical non-identifiability is harder to resolve even if general methods have been proposed [1]. Here, we chose the NF-κB regulatory pathway, as recent works point out that the known detailed models of this very important pathway are not identifiable [2–4]. We reduce the original 15-variable NF-κB model developed by Lipniacki et al. [5] in a way that it preserves two negative feedback loops

**Funding:** This research was supported by National Science Centre (Poland) grant 2018/29/B/NZ2/00668 and Norwegian Financial Mechanism GRIEG-1 grant (operated by the National Science Centre, Poland) 2019/34/H/NZ6/00699 to TL. During the initial stage of the project IK was supported by the European Union's Horizon 2020 research and innovation program under the Marie Sklodowska-Curie Grant Agreement No 661650. IK thanks Vienna University of Technology for support and hospitality. The funders had no role in study design, data collection and analysis, decision to publish, or preparation of the manuscript.

controlling NF-$\kappa$B activity and mediated by I$\kappa$B$\alpha$ and A20 (in contrast to ealier even simpler models [6–8]), but makes the model structurally and practically identifiable. Our approach is not automatic, but rather modeler driven and thus differs from the algorithmic methods tested previously on the same model [2, 9, 10] or its later variant [11].

NF-$\kappa$B is an important transcription factor controlling expression of numerous genes regulating the innate immune response. It is activated in response to TNF or IL1 cytokine stimulation, as well as viral RNA or bacterial LPS [12]. In resting cells, NF-$\kappa$B forms inactive cytoplasmic complexes mainly with its primary inhibitor I$\kappa$B$\alpha$. The signals activating NF-$\kappa$B converge on the cytoplasmic I$\kappa$B kinase, IKK. Activated IKK phosphorylates I$\kappa$B$\alpha$, inducing its ubiquitination and rapid proteasomal degradation. I$\kappa$B$\alpha$ degradation enables NF-$\kappa$B translocation to the nucleus, where NF-$\kappa$B upregulates transcription of numerous genes including genes of its two inhibitors, I$\kappa$B$\alpha$ and A20 [5, 13]. The newly synthesized I$\kappa$B$\alpha$ translocates to the nucleus, binds NF-$\kappa$B and their complex is transported to the cytoplasm, which completes the I$\kappa$B$\alpha$ feedback loop. A second level of negative regulation of NF-$\kappa$B involves A20 that attenuates IKK activity by promoting its transformation into the inactive overphosphorylated form. In this way A20 allows for accumulation of the primary NF-$\kappa$B inhibitor I$\kappa$B$\alpha$ [5, 14]. A20 is also an inhibitor of kinase TBK1, which activates IRF3, the other critical transcription factor regulating the innate immune response [15]. As a consequence, A20 regulates the crosstalk between NF-$\kappa$B and IRF3, which are both activated upon recognition of pathogens and are both necessary for the synthesis of interferons, produced to warn surrounding cells about infection [12].

Modeling efforts that started two decades ago resulted in the formulation of detailed biochemical models of the NF-$\kappa$B signaling pathway [5, 16], reviewed in [17, 18]. Models, supported by live single-cell experiments [19, 20], elucidated the role of I$\kappa$B$\alpha$- and A20-mediated negative feedbacks loops in promoting oscillations observed in several cell lines, including SK-N-AS and MEF 3T3. These two feedback loops play also an important role in shaping the pulsatile NF-$\kappa$B response to the pulsatile TNF stimulation [21, 22]; as demonstrated, A20-deficient cells show no NF-$\kappa$B oscillations to tonic TNF stimulation [14] and exhibit prolonged NF-$\kappa$B activation in response to a pulse of TNF [23]. Individual cells have the ability to respond to the pulsatile TNF stimulation with synchronous NF-$\kappa$B pulses, which result in a high expression of NF-$\kappa$B-regulated genes [20, 24]; interestingly, for certain TNF stimulation periods, due to noise the cells may switch between two entrainment modes of different frequencies [25]. Pulsatile stimulation was also shown to transmit more information than the continuous one, with the rate of order 1 bit/hour [26].

The main goal of this study is to obtain an accurate, practically identifiable model, and suggest a reasonable experiment allowing for a determination of all parameters with a satisfactory accuracy. We found that a simple protocol with a continuous 2 hour long TNF stimulation followed by a 10 hour long TNF washout phase assures not only practical identifiability of the reduced model, but also allows to determine parameters with accuracy comparable to the more complex pulsatile protocols. We evaluated parameter identifiability using sensitivity analysis techniques [27–30] finding the parameters that may be determined with the lowest accuracy. We verified practical identifiability performing Monte Carlo simulations [31]. Following the standard Monte Carlo procedure, for each of three different measurement noise levels, we generated 50 different trajectories. Next the model was fitted to such simulated experimental trajectories and 50 parameter estimates were obtained. The results indicate that all parameters can be estimated based on noisy data.

Finally, we showed that the reduced model has ability to produce qualitatively distinct dynamical responses to continuous TNF signal. Depending on the assumed parameters it can exhibit a pulse like response, damped oscillations, or sustained oscillations, i.e., the essential

characteristics of the negative feedback system. Importantly, our results suggest that other regulatory pathways models can be transformed to identifiable, while retaining their key dynamical features.

## Results

### Original two-feedbacks model and its reduction

The model of Lipniacki et al. [5] incorporates two negative feedback loops, one mediated by I$\kappa$B$\alpha$ directly binding NF-$\kappa$B and sequestering it in the cytoplasm, and the other involving the inhibition of IKK (kinase responsible for phosphorylation of I$\kappa$B$\alpha$ targeting it for proteasomal degradation) by the protein A20 (see Fig 1A). The dynamics of 15-model variables has been described with mass action kinetics. For wild type (WT) cells, the dynamics of main variables in the Lipniacki et al. 2004 model [5] qualitatively follows the dynamics of the earlier Hoffmann et al. 2002 model [16], see S1 Fig and S1 Text for details of the Hoffmann et al. 2002 model simulations. The model comparison involves tonic TNF stimulation, as well as pulsatile TNF stimulations following experimental protocols published in [21, 22] after the both models were developed. We may notice that the Lipniacki et al. 2004 model [5] satisfactorily reproduces free nuclear NF-$\kappa$B trajectory for the protocols with three 5-min TNF pulses separated by 60 min, 100 min or 200 min intervals studied by Ashall et al. ([21], see Fig 2A) as well as for pulsatile protocols studied by Zambrano et al. ([22], see Fig 1C—a series of 45min TNF pulses separated by 45 min intervals and Fig 3, S1D Fig—a series of 22.5 min TNF pulses separated by 22.5 min intervals). As observed earlier in [24, 25] the NF-$\kappa$B system can be entrained to the pulsatile stimulation. The important differences between these two models are as follows: The Hoffmann et al. 2002 model accounts for 3 I$\kappa$B isoforms, whereas the Lipniacki et al. 2004 model replaces them by a single dominant I$\kappa$B isoform, I$\kappa$B$\alpha$. In contrast to the Hoffmann et al. 2002 model [16], the Lipniacki et al. 2004 model [5] accounts for A20, an NF-$\kappa$B-inducible inhibitor of NF-$\kappa$B. A20 attenuates activity of IKK, mediating another crucial negative feedback loop. As a consequence, the latter model can reproduce trajectories of both WT and A20-deficient cells, as demonstrated in the original publication [5].

Different, to the Lipniacki et al. 2004 model [5], A20 feedback structures were proposed by Ashall et al. 2009 [21], and Murakawa et al. 2015 [32], and the dynamical consequences of these three A20 feedback structures were discussed in-depth by Mothes et al. [33], see also S1 Text for mathematical details of the Ashall et al. and Murakawa et al. models. In brief, the Lipniacki et al. 2004 model assumes that upon TNF stimulation IKK transits from its resting form, IKKn, to active form IKKa. The A20 protein, in turn, promotes transition of IKKa to inactive form, IKKi. Consequently, in WT cells in response to the tonic stimulation, IKKa exhibits a short pulse with a low tail (S2 Fig), while in A20-deficient cells, the tail is higher, because the transition from IKKa to IKKi proceeds at a lower rate (S3 Fig). Ashall et al. assumed also existence of these three forms of IKK, but proposed that A20 instead of promoting IKKa to IKKi transition, slows down the transition from IKKi to IKKn. The consequence of this assumption is similar: in WT cells, IKKn is more depleted than in A20-deficient cells, and consequently the tail of IKKa is lower (compare S2 and S3 Figs). These two models produce IKKa profiles in agreement with the experiment conducted by Lee et al., 2000 [14], that motivated incorporation of A20 feedback to NF-$\kappa$B modeling. These experimental data ([14], see Fig 3E therein) indicate a pronounced peak of IKKa 10 minutes after the beginning of TNF stimulation followed by a tail of IKK activity—low for WT cells and higher for A20-deficient cells. A substantially different IKK-A20 module structure was proposed in the Murakawa et al. 2015 model [32], which accounts only for one (active) form of IKK. Accumulation of active IKK is slowed down by A20, while the depletion term of active IKK is constant. Consequently,

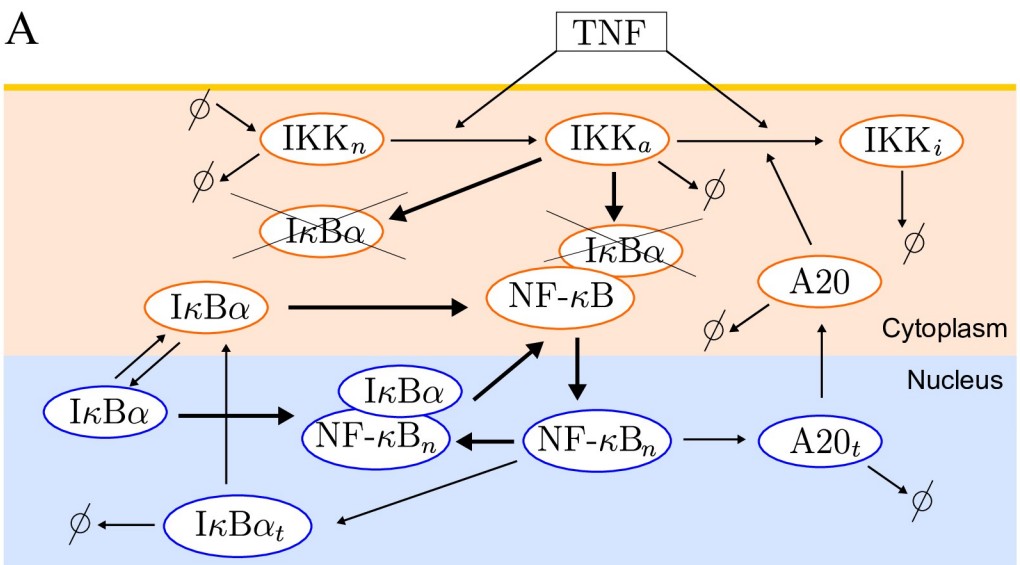

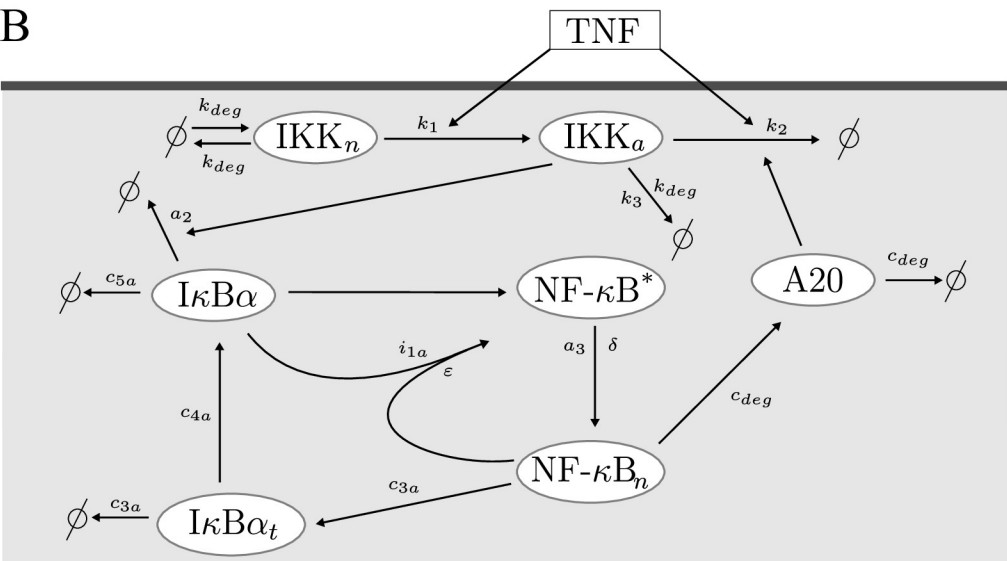

**Fig 1. NF-κB pathway diagrams.** (A) Diagram of the 15-variable original model. The short lasting cytoplasmic complexes (IKK|IκBα) and (IKK|NF-κB|IκBα) are not depicted for the sake of simplicity. The bold arrows represent the fast dynamics. (B) Diagram of the reduced 6-variable model. NF-κB* denoting cytoplasmic (NF-κB|IκBα) complexes is not associated with the separate equation, since the amount of total NF-κB remains constant: $NF\kappa B_n + NF\kappa B^* = 1$.

in WT cells, due to accumulation of A20, active IKK also exhibit a peak (S2 Fig). However, in A20-deficient cells, the level of active IKK grows monotonically to some asymptotic value (S3 Fig), which is in stark contrast to Lee et al., 2000 data [14]. For WT cells, the Murakawa et al. 2015 model [32] (as well as the Hoffmann et al. 2002 model [16]) fails to respond to 3 TNF pulses separated by 200 min intervals (S1 and S2 Figs) that as demonstrated by Ashall et al. 2009 [21] induce 3 peaks of NF-κB activity; this discrepancy is possibly a consequence of models parametrization not A20-IKK module structure, or lack of this module.

Starting from the Lipniacki et al. 2004 model [5], hereafter referred to as the original model, we retained the core of the two-feedback network and constructed a reduced model still

exhibiting qualitatively similar kinetics. The detailed description of all reduction steps is given in S2 Text.

The key reduction steps can be summarized as follows:

1) We have eliminated the short-lived complexes (IKKa|IκBα) and (IKKa|IκBα|NF-κB). Instead we equate inflows and outflows rates from these two complexes.

2) We have eliminated cytoplasmic NF-κB (because it either rapidly binds to cytoplasmic IκBα or translocates to the nucleus), nuclear IκBα (because it rapidly binds to nuclear NF-κB), and the nuclear complexes of (IκBα|NF-κB), because they rapidly translocate to the cytoplasm.

3) We have eliminated A20 mRNA, whose formation is a step in A20 protein synthesis. However, we retained IκBα mRNA, because we found that the time delay associated with that intermediate step (in contrast to the delay introduced by A20 mRNA) is important for the observed oscillatory behavior.

4) The model was transformed to the non-dimensional form. As a consequence of non-dimensionalization degradation coefficients for IKKn, A20, and IκBα are equal to the corresponding synthesis coefficients (see Table 1).

The remaining variables are two forms of cytoplasmic IKK (IKKn and IKKa), free nuclear NF-κB, proteins A20 and IκBα, and IκBα transcript. The resulting model is depicted in Fig 1B and described by the following system of kinetic equations based on mass action or Michaelis–Menten kinetics.

IKK in its neutral state IKKn. The three terms stand consecutively for IKKn protein synthesis, degradation (with the same coefficient), transition to active IKKa form in response to TNF stimulation ($T_R = 1$ for TNF ON and $T_R = 0$ for TNF OFF)

$$IKKn' = k_{deg} - k_{deg}IKKn - T_R k_1 IKKn. \qquad (1)$$

IKK in the active state IKKa. The terms stand consecutively for transition from IKKn form and removal of IKKa (associated with spontaneous transition to inactive form, degradation,

**Table 1. Parameter values for the reduced and reduced fitted model.**

| Description | Parameter | Pre-fitted value | Fitted value | Unit |
|---|---|---|---|---|
| Synthesis and degradation of IKKn | $k_{deg}$ | 0.000125 | 0.000107 | $s^{-1}$ |
| Degradation of IKKa | | | | |
| Activation of IKKn caused by TNF | $k_1$ | 0.0025 | 0.00195 | $s^{-1}$ |
| Spontaneous inactivation of IKKa | $k_3$ | 0.0015 | 0.00145 | $s^{-1}$ |
| Inactivation of IKKa induced by A20 | $k_2$ | 0.0625 | 0.0357 | $s^{-1}$ |
| IKK (IκBα|NF-κB) association | $a_3$ | 0.2 | 0.0946 | $s^{-1}$ |
| leading to IκBα degradation | | | | |
| Michaelis-type constant | $\delta$ | 0.0833 | 0.108 | |
| Michaelis-type constant | $\varepsilon$ | 0.0167 | 0.0428 | |
| Synthesis and degradation of A20 | $c_{deg}$ | 0.000171 | 0.000106 | $s^{-1}$ |
| IκBα translation | $c_{4a}$ | 0.0031 | 0.00313 | $s^{-1}$ |
| IκBα degradation induced by IKKa | $a_2$ | 0.04 | 0.0763 | $s^{-1}$ |
| IκBα degradation | $c_{5a}$ | 0.0001 | 0.0000578 | $s^{-1}$ |
| IκBα nuclear import leading to | $i_{1a}$ | 0.001 | 0.000595 | $s^{-1}$ |
| NF-κB removal from the nucleus | | | | |
| IκBα synthesis and degradation of IκBα mRNA | $c_{3a}$ | 0.0004 | 0.000372 | $s^{-1}$ |

and transition to inactive form promoted by A20 and TNF)

$$IKKa' = T_R k_1 IKKn - (k_3 + k_{deg} + T_R k_2 A20) IKKa. \tag{2}$$

Free nuclear NF-κB. The first term stands for appearance of nuclear NF-κB due to degradation of IκBα in (NF-κB|IκBα) complexes (whose level is equal to $1 - NFκB_n$). The IκBα degradation is proportional to IKKa (which phosphorylates IκBα and targets it for rapid degradation). The appearance of active NF-κB is blocked by free IκBα (that may bind it before its nuclear translocation). The second term describes removal of nuclear NF-κB due to rapid binding with IκBα. This process is regulated by IκBα nuclear translocation with rate $i_{1a}$

$$NFκB_n' = a_3 IKKa \cdot (1 - NFκB_n) \frac{\delta}{IκBα + \delta} - i_{1a} IκBα \frac{NFκB_n}{NFκB_n + \varepsilon}. \tag{3}$$

A20 protein. The two terms in the fourth equation describe A20 protein synthesis regulated by NF-κB and its degradation, respectively

$$A20' = c_{deg} NFκB_n - c_{deg} A20. \tag{4}$$

Free cytoplasmic IκBα protein. The terms stand consecutively for IκBα protein translation, spontaneous degradation, IKKa regulated degradation. The fourth term arises due to reduction of several process: it describes depletion of free IκBα due to its binding with free NF-κB that in turn arises due to IKK induced degradation of IκBα bounded to NF-κB. The fifth term is equal to the second term in the Eq (3) and describes depletion of free IκBα due to binding to nuclear NF-κB

$$\begin{aligned} IκBα' &= c_{4a} IκBα_t - c_{5a} IκBα - a_2 IKKa \cdot IκBα \\ &\quad - a_3 IKKa (1 - NFκB_n) \frac{IκBα}{IκBα + \delta} - i_{1a} IκBα \frac{NFκB_n}{NFκB_n + \varepsilon}. \end{aligned} \tag{5}$$

IκBα transcript. The terms in the sixth equation describe IκBα mRNA synthesis regulated by NF-κB and its degradation, respectively

$$IκBα_t' = c_{3a} NFκB_n - c_{3a} IκBα_t. \tag{6}$$

Possibly, the reduced model, consisting of 6 variables is the smallest one that can account for two regulatory feedback loops mediated by IκBα, a direct NF-κB inhibitor, and A20 inhibitor of IKK (which in its active form directs IκBα for degradation). The model accounts for IκBα mRNA, which is needed to assure a time delay between NF-κB activation and IκBα protein synthesis. This delay enables oscillatory responses. We were able to skip A20 mRNA, as A20 regulates NF-κB in an indirect way, via IKK and IκBα, which assures a sufficient time delay. Nevertheless, a model accounting also for A20 mRNA seems to be a reasonable alternative, although it would require A20 mRNA measurements to constrain respective parameters.

## Refitting the reduced model to the original one

To verify that the reduced model may accurately replace the original one, we analyze responses of both models to an *in silico* experiment involving six TNF stimulation protocols (the tonic stimulation and 5 pulsatile protocols defined in S1 Table), which were used in published experiments [21, 22], see Fig 2. In this numerical experiment (hereafter referred to as the combination experiment) we compared numerical trajectories of WT and A20-knocked out (A20 KO) cells of the 5 main variables (IKKa, NF-κB, A20, total IκBα, and IκBα mRNA), generated by the original model, the reduced model and the reduced model fitted to the original one. The

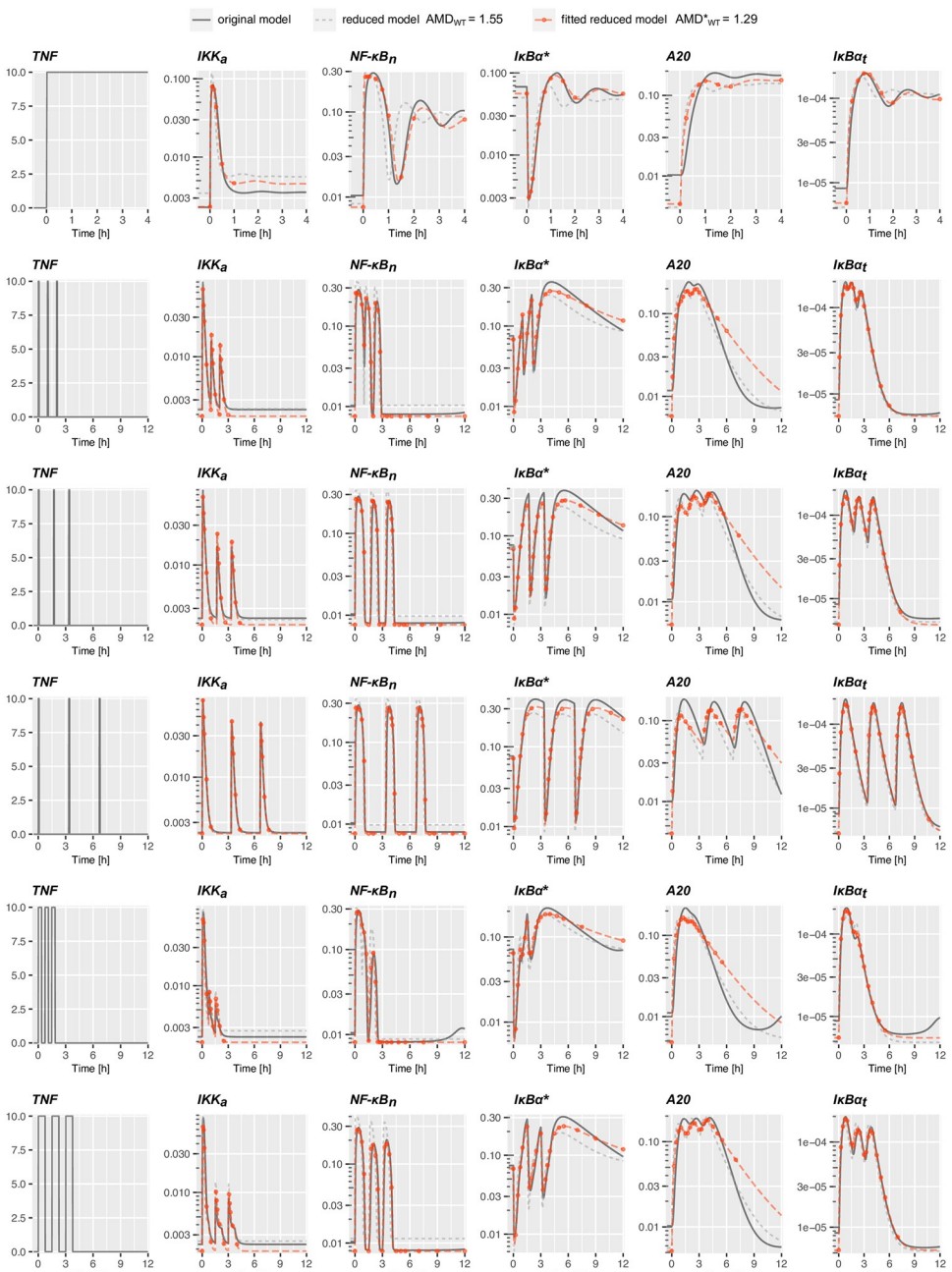

**Fig 2. Dynamics of the original, reduced and reduced fitted models in the combination experiment defined in S1 Table.** Red dots indicate time points from which the nuclear NF-κB and total IκBα protein *in silico* measurements are used for fitting the reduced model to the original one. The time points for the remaining variables are given in S1 Table.

combination experiment contains 914 independent data points (defined in S1 Table, see also S2 Table). The reduced model has been fitted to the original one using PʏBɪoNᴇᴛFɪᴛ [34] minimizing squared distances of log-transformed solutions in time points given in S1 Table (see Methods for details). The applied log-transformation follows from the assumption that only the relative (not the absolute) values of all variables are measured. The accuracy of the reduced

model and the fitted reduced model with respect to the original one is assessed by the Average Multiplicative Distance (AMD)—see Methods for details. For WT cells the average distance of the reduced model and the original one is $AMD_{WT} = 1.55$, whereas the distance of the reduced fitted model and the original one is $AMD^*_{WT} = 1.29$ (implying about 30% differences between the two models trajectories). For A20 KO cells these distances are smaller, equal respectively, $AMD_{A20KO} = 1.28$ and $AMD^*_{A20KO} = 1.16$. In Fig 2 we show the NF-$\kappa$B trajectories for the six simulated protocols for the three models; the corresponding trajectories for A20 KO cells are shown in S4 Fig. Both figures and the low AMD values between the fitted reduced model and the original one indicate that the reduced model after refitting may accurately replace the original model reproducing trajectories of the 5 main model variables in response to various TNF stimulations, and as such may reproduce all experimental data reproduces by the original model.

In Table 1 we provide numerical values of the 13 parameters of the reduced model, before and after refitting them to the original one. The differences between these two sets of parameters are significant, confirming that refitting is an important step after model reduction. With respect to the original model there are two novel parameters: $\delta$ and $\varepsilon$. Parameter $\delta$ is proportional to NF-$\kappa$B nuclear import and inversely proportional to I$\kappa$B$\alpha$—NF-$\kappa$B binding. Parameter $\varepsilon$ is proportional to I$\kappa$B$\alpha$ nuclear export and also inversely proportional to I$\kappa$B$\alpha$ and NF-$\kappa$B binding. Both $\delta$ and $\varepsilon$ are smaller than 1, which reflects the original model assumption that the I$\kappa$B$\alpha$—NF-$\kappa$B binding is a faster process than nuclear-cytoplasmic trafficking.

The original model has been constrained based on experiments on mouse embryonic fibroblasts and further verified also on experiments on SK-N-AS cells [21]. Since the other cell lines may be characterized by different parameters, it is important to verify whether the reduced model can be fitted to the original one also for different sets of parameters. Because the reduced model relies on approximations that are based on the existence of fast processes, one can expect that the ability to closely reproduce the original model might be limited to ranges of the original parameter set for which these processes can still be considered fast. In Fig 3 we have randomly selected five parameter sets of the original model varying the original parameters at most three-fold from their original values (see S3 Table) such that the resulting model trajectories vary significantly. Next, the reduced model has been fitted to these five new variants of the original model (see S4 Table). The comparison of the nuclear NF-$\kappa$B dynamics of the original and reduced models is shown in Fig 3. For each parameter set the corresponding $AMD^*_{WT}$ and $AMD^*_{A20KO}$ are computed as previously based on the combination experiment. The obtained fits have $AMD^*_{WT}$ values in range 1.22—1.41, and $AMD^*_{A20KO}$ in range 1.13—1.33 (The NF-$\kappa$B trajectories for A20 KO cells for the combination experiment are provided in S5 Fig). The small AMD values confirm the ability of the reduced model to reproduce (after the parameter refitting) the original one also for different sets of parameters (varied in relatively broad range).

In summary, the reduced model may reproduce the NF-$\kappa$B system dynamics (for WT and A20 KO cells) of the original model both for the original as well as perturbed parameters. Therefore, it represents a good approximation for studying the behavior of the NF-$\kappa$B signaling pathway. Importantly, we will show that thanks to a smaller number of parameters, the reduced model is, in the contrary to the original one, identifiable.

## Structural identifiability analysis of the original and reduced model

In this section we perform the linear identifiability analysis based on the sensitivity matrix for the original as well as of the reduced model fitted to the original one, hereafter referred as reduced model. In order to demonstrate non-identifiability of the original model based on

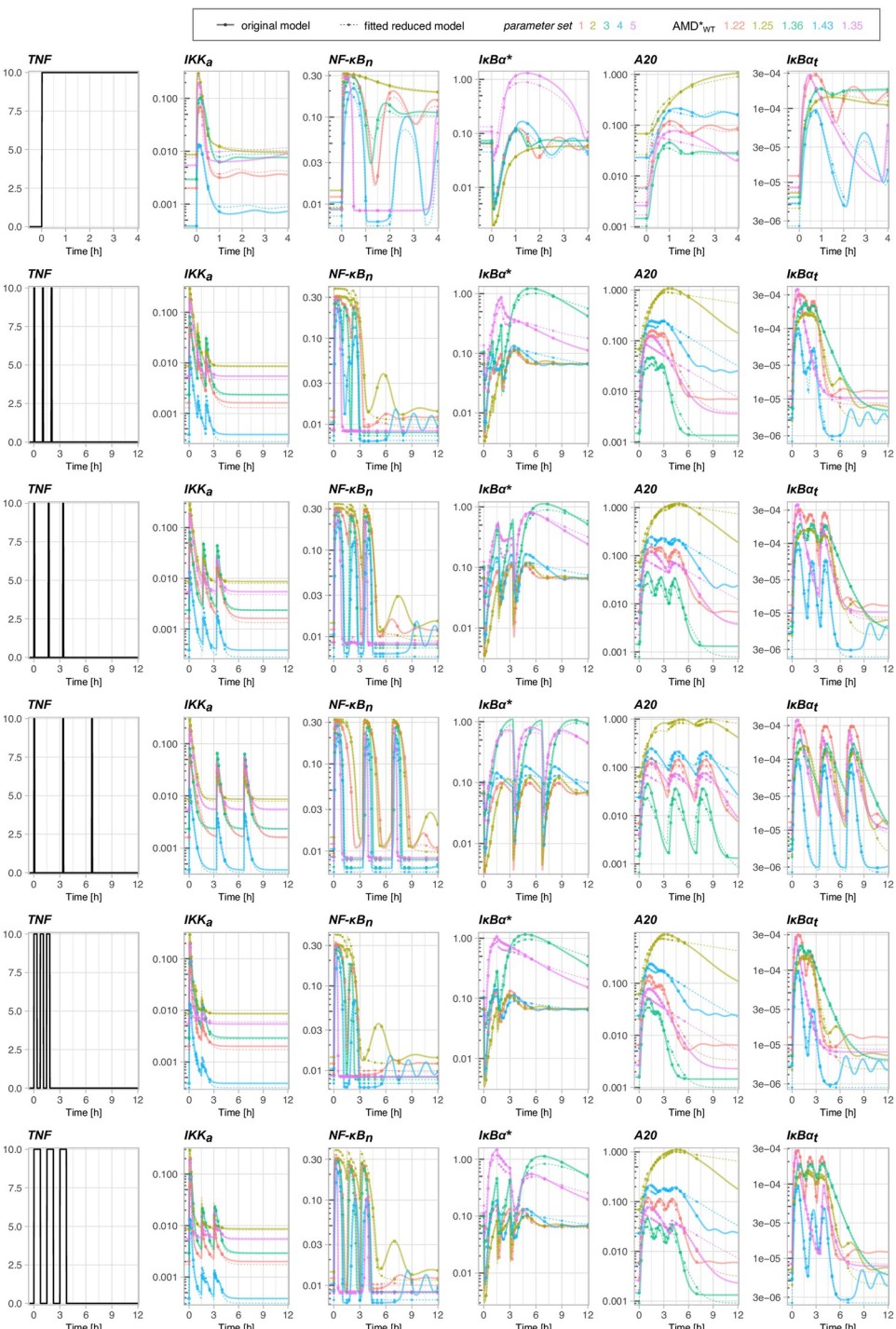

**Fig 3. Simulations of the original model for five different sets of parameters and the corresponding reduced model with refitted parameter values (see S3 and S4 Tables).** Simulations are performed for the combination experiment defined in S1 Table. For each parameter set the corresponding $AMD^*_{WT}$ is computed based on trajectories of the 5 main model variables. Each color corresponds to a different parameter set; for each parameter set, a continuous line shows the original model trajectory, while a thinner dashed line shows the corresponding trajectory of the reduced model with refitted parameters.

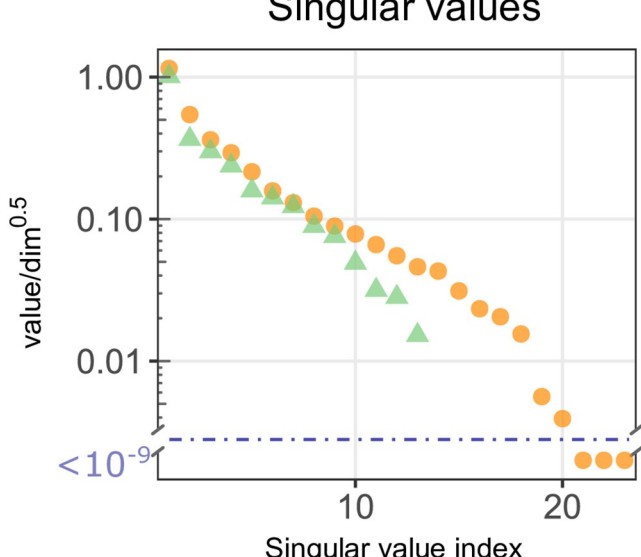

**Fig 4. Structural identifiability analysis of the reduced and original models.** Singular values of the sensitivity matrices *S* for the original and reduced models. Graphs show all 23 scaled singular values obtained (for the combination experiment) for the original model vs. 13 singular values obtained for the reduced model. Singular values, arranged in descending order, in the original model reveal a clear gap, indicating that the corresponding sensitivity matrix is rank deficient and hence the original model is structurally non-identifiable.

experiments that have been performed so far [21, 22], we consider the combination experiment (consisting of tonic and 5 pulsatile protocols defined in S1 Table). Based on this huge hypothetical experiment (with 914 independent data points), we calculate the sensitivity matrix *S* numerically differentiating log-transformed normalized observables with respect to the log-transformed parameters (as detailed in Methods) for the original model as well as for the reduced model. Based on singular value decomposition of sensitivity matrices we calculate singular values for the original and reduced models. The calculated singular values are scaled by dividing by the squared root of sensitivity vectors dimension (*dim*), which assures that they are unchanged by simple repetition of the experiment doubling *dim*. It should be noticed that, because of data normalization, the dimension of sensitivity vectors is smaller than the total number of data points—for example a series of three data points gives only two independent numbers. As shown in Fig 4 for the original model the scaled singular values (hereafter referred as singular values), arranged in descending order, reveal a clear gap indicating the corresponding sensitivity matrix is rank deficient and hence the original model is structurally non-identifiable. In contrast, for the reduced model all singular values are greater than 0.01 indicating structural identifiability.

### Detailed linear identifiability analysis of the reduced model

In Fig 5 we evaluate which experimental protocols, defined in Table 2 and S1 Table, allow for the most accurate parameter identifiability. In Fig 5A we show that three smallest singular values for all considered protocols are greater than $10^{-3}$, indicating that each of protocols renders

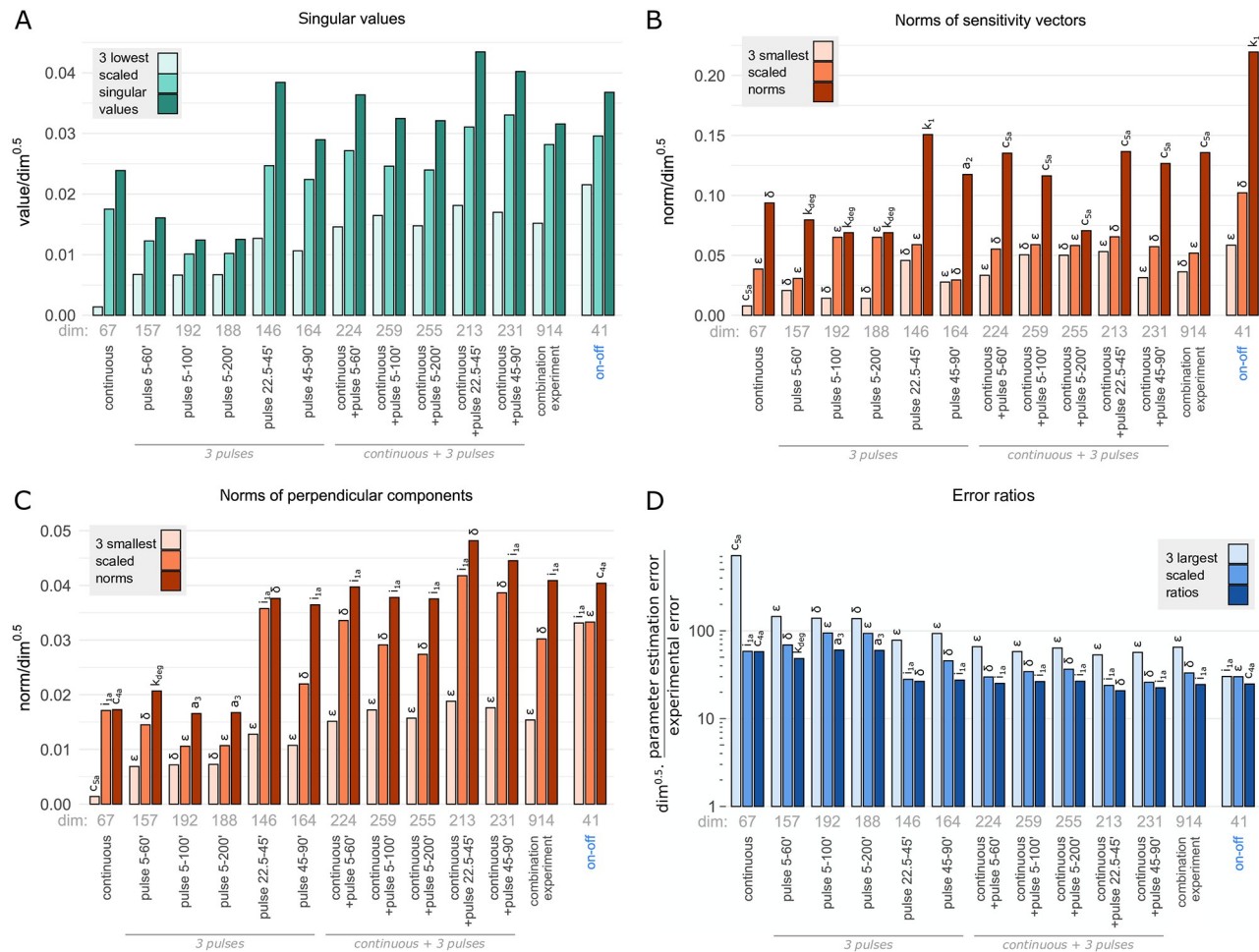

**Fig 5. Linear identifiability analysis of the reduced model.** (A) Comparison of the three smallest scaled singular values computed for different protocols. (B) Smallest scaled norms of sensitivity vectors $||S_j||$. (C) Three smallest scaled norms of perpendicular components $||S_j^\perp||$. (D) Three largest scaled ratios of the parameter estimation error to experimental error $R_j = \ln(\sigma_{linear,j})/\ln(\sigma_{data})$. The combination experiment includes the continuous and 5 pulsatile protocols. Scaling with $\sqrt{dim}$ allows to compare the protocols having a different dimension of sensitivity vectors.

the reduced model structurally identifiable. We should, however, notice that for the continuous protocol the identifiability is poorest with the smallest singular value equal to 0.0014. For all pulsatile protocols the smallest singular value exceeds 0.007, and when these protocols are combined with the continuous protocol, the smallest singular value rises above 0.014.

**Table 2. Summary of the on-off protocol of the NF-κB system in WT and A20 KO cells used for *in silico* experiments.** A simple protocol with a continuous TNF stimulation (0–120min) followed by a TNF washout (120–720min) for which measured variables are: IKKa, nuclear NF-κB, A20, total IκBα protein denoted by IκBα* (IκBα* = IκBα + 1 − NFκB), and IκBα mRNA. For each variable the measurement times used for the sensitivity-based identifiability analysis and reduced model fitting are given.

| on-off | TNF *ON* 0–120 min, TNF *OFF* 120–720 min |
|---|---|
| IKKa | 0, 5, 30 min |
| NF-κB | 0, 5, 30, 60, 90, 120, 150, 180 min |
| IκBα* | 0, 5, 30, 60, 120, 180, 300, 720 min |
| A20, IκBα$_t$ | 0, 30, 60, 300 min |

Interestingly, the simple on–off protocol (proposed in this study, see Table 2) with the smallest number of data points outperforms all remaining protocols having the smallest singular value equal to 0.022. It is important to notice that the comparison of protocols in Fig 5A is based on scaled singular values, i.e., divided by the squared root of sensitivity vectors dimension. Obviously, the combination experiment having 914 independent data points has greater singular values, than the on–off protocol having only 41 independent data points. The greater scaled singular values of the on–off experiment imply, however, that in order to better identify parameters it pays to repeat the on–off protocol $914/41 \approx 22$ times than to do the combination experiment. For the same reason analyzing the identifiability of the model parameters in Figs 5C, 5D and 6 we used the $\sqrt{dim}$ scaled values.

After verifying that the reduced model is structurally identifiable, we further investigate the sensitivity matrix $S$ to see which model parameters are the least sensitive and least identifiable for a given protocol. Fig 5B illustrates the three smallest (scaled) norms corresponding to the least sensitive parameters. It is worth to notice that the parameters $\varepsilon$ and $\delta$ appear as two out of three least sensitive parameters for all protocol sets. For the continuous protocol the scaled norm of the sensitivity vector for the parameter $c_{5a}$ (governing IκBα degradation) is the smallest (equal to 0.0078) and significantly higher in the on–off protocol (equal to 0.28), which allows to trace IκBα degradation.

In Fig 5C we show the scaled norms of perpendicular parts of sensitivity vectors $||S_j^\perp||$ that provide a good measure of identifiability of the associated parameters. A small perpendicular part implies that a change of the associated parameter can be nearly fully compensated by changes of remaining parameters, rendering the considered parameter poorly identifiable. As expected for the continuous protocol the parameter $c_{5a}$ is least identifiable, however, for the pulsatile protocols the least identifiable are the parameters $\delta$ and $\varepsilon$.

In Fig 5D we show the three largest scaled ratios of the parameter estimation error to the experimental error. The calculation is based on assumption that all experimental points have lognormally distributed errors with the same geometric standard deviation $\sigma_{data}$. Then the parameter estimation error to the experimental error ratio is equal to $\ln(\sigma_{linear,j})/\ln(\sigma_{data})$, where $\sigma_{linear,j}$ is the corresponding geometric standard deviation of parameter estimation error; the subscript 'linear' stands for the method of calculation using sensitivity matrix $S$ (see Methods for details). Not surprisingly, for each protocol, the three largest error ratios correspond to the three smallest norms of perpendicular parts $||S_j^{\perp|}||$ confirming that these parameters are the least identifiable. Fig 5C and 5D, show that the on–off protocol out-competes the remaining protocols or their combinations. For this protocol the least identifiable parameter, $i_{1a}$, is better identifiable than the least identifiable parameter for any of remaining protocols.

In Fig 6 we focus the identifiability analysis on three particular experiments, i.e., the continuous protocol, the combination experiment, and the on–off protocol. Fig 6A shows that the simple on–off protocol, out-competes remaining two protocols giving the higher scaled singular values. When the norms of sensitivity vectors Fig 6B, or their perpendicular components Fig 6C are juxtaposed, the on–off protocol is comparable to the combination experiment. In Fig 6B and 6C, parameters stand in the growing order of respectively norms of sensitivity vectors and their perpendicular components for the on–off protocol. We may notice that the sensitive parameters like $i_{1a}$ (i.e., having a high norm of sensitivity vector) maybe poorly identifiable, i.e., may have a small perpendicular component (which allows to partially compensate a change of such parameters by respective changes of remaining parameters). A comparison of Fig 6C and 6D (with palindromic order of parameters), in turn confirms that parameters with small perpendicular components have a large parameter estimation error to experimental error ratio (and thus may be poorly identifiable based on noisy data).

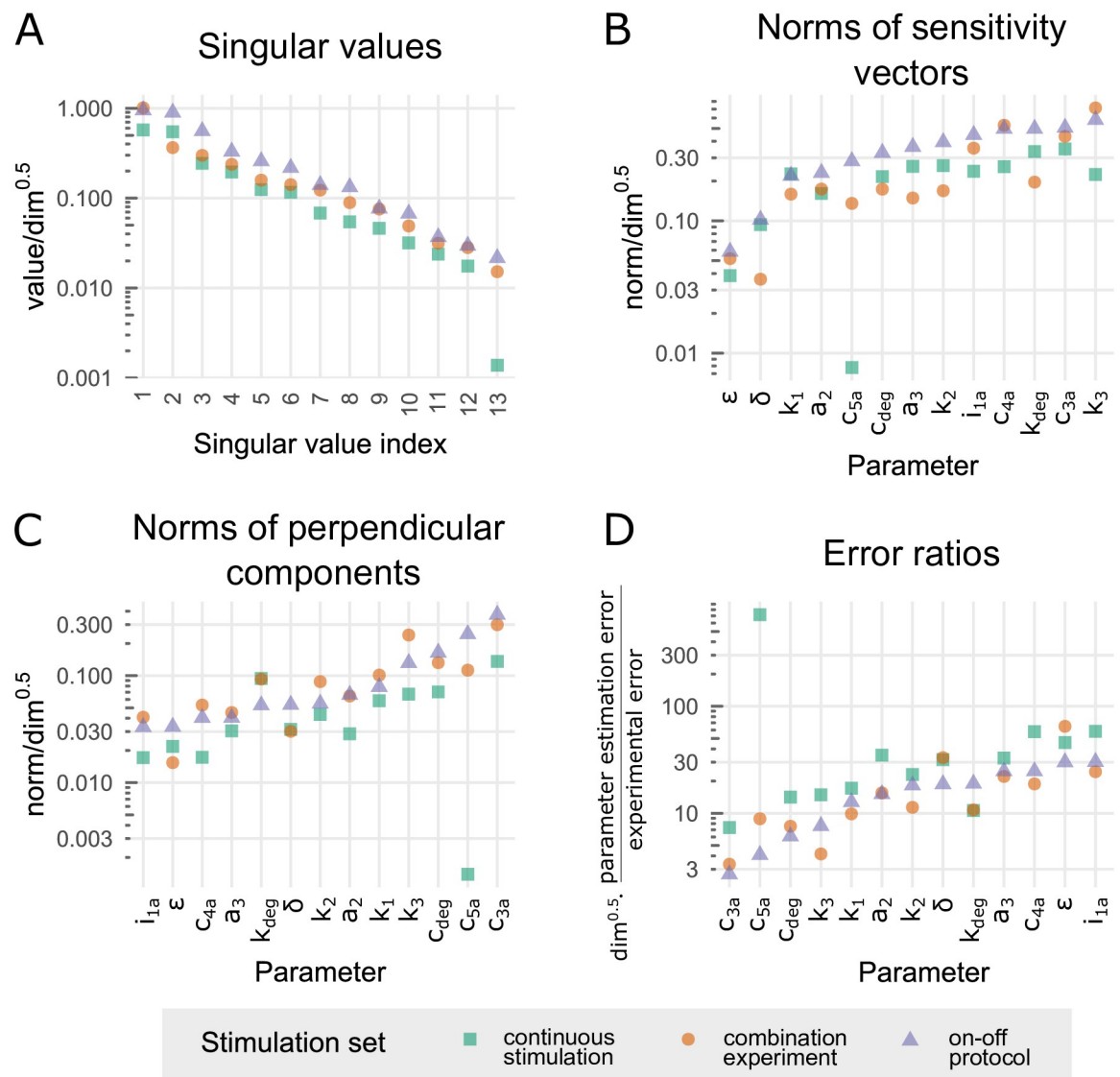

**Fig 6. Identifiability analysis of the reduced model—comparison of the protocols: Continuous, combination experiment and on–off.** (A) Scaled singular values. (B) Scaled norms of sensitivity vectors $||S_j||$. (C) Scaled norms of perpendicular components $||S_j^\perp||$. (D) Scaled ratios of parameter estimation error to experimental error $\ln(\sigma_{linear,j})/\ln(\sigma_{data})$. Scaling with $\sqrt{dim}$ allows for a comparison of the three protocols having a different dimension of sensitivity vectors.

To sum up, according to our sensitivity-based linear analysis all model parameters are identifiable, however, the continuous protocol results with very poor identifiability of $c_{5a}$ (governing the IκBα degradation). The simple on-off protocol results in comparable or better parameters identifiability than the more complex protocols including the combination experiment. The idea behind the on-off protocol is such that a 2 hour-long TNF stimulation phase suffices for a rise of all variables (except IKKn, which decreases in response to TNF), and a 10 hour-long washout phase allows for their substantial decrease S6 Fig. Such an experiment, surprisingly not often conducted, allows to determine both forward and backward rates. The time points for fitting are distributed more densely in time intervals in which given variables change more rapidly, which typically means higher sensitivity to parameters. However, to make the

protocol experimentally feasible, the number of time points is limited (i.e., time points for different variables partially overlap).

## Practical identifiability analysis of the reduced model based on Monte Carlo simulations

In this section we verify the practical identifiability of model parameters with help of Monte Carlo simulation [31, 35]. We use the fitted reduced model to simulate the experimental trajectory for the on-off protocol (S6 Fig) Next, we use 50 *in silico* simulated measurements (generated by the reduced model as defined in Table 2) from the on-off protocol trajectory to refit the model again (using stochastic fitting algorithm; see Methods). Alternatively, we perturb randomly these measurements before refitting. In this latter procedure we mimic the experimental errors [36]. More precisely, we draw values of measurements from the lognormal distribution, Lognormal($\mu$, $\sigma^2$), with median $\exp(\mu)$ equal to the unperturbed value, and $\sigma = \ln(\sigma_{data})$, where $\sigma_{data}$ is the assumed geometric standard deviation (sometimes called a multiplicative standard deviation) of measurement values. In Fig 7 we chose three values of $\sigma_{data}$, equal to 1.1, 1.2 and 1.3, and for each $\sigma_{data}$ we draw $k = 50$ sets of measurements. For each set of measurements we refit the 13 model parameters. For Fig 7 we select 6 parameters with the highest geometric standard deviation and project $k$ refitted sets of these parameters on the $5 \times 6/2 = 15$ respective planes. We also show the marginal distribution for each of these 6 parameters. The results for the remaining 7 (better identifiable) parameters are illustrated in S7 Fig. Of note 50 *in silico* simulated measurements in the on-off protocol correspond to 41 independent data points, because $n$ time points of any time series give $n - 1$ independent values due to scaling by the geometric average of the series. In the on-off protocol there are 5 series for WT cells for 5 variables, IKKa, NF-κB, A20, total IκBα, and mRNA IκBα, and 4 series for A20 KO cells for the same variables excluding A20. As a consequence the number of independent data points is by 9 smaller than the number of measurements, see S2 Table.

The obtained parameter scatter plots for $\sigma_{data} = 1.3$ are confronted with 75% confidence ellipses obtained from the linear analysis for the same $\sigma_{data}$, showing that parameter uncertainty determined by the linear analysis is comparable to that of determined by Monte Carlo simulations. Fig 7A shows that due to the use of stochastic fitting algorithm, even when unperturbed measurements are used for $k$ independent fittings, the obtained parameters differ from the original ones and correspondingly, their geometric standard deviations $\sigma_{carlo,j}$ are greater than 1, see Fig 7B. As shown in Fig 7B for all parameters $\sigma_{carlo,j}$ grows with $\sigma_{data}$. For most of the parameters the value of $\sigma_{carlo,j}$ obtained for $\sigma_{data} = 1.3$ is comparable to the value of $\sigma_{linear,j}$ obtained in the linear noise approximation also for $\sigma_{data} = 1.3$, while for some of them (e.g. $\delta$) is markedly larger, and a bit surprisingly for some of them (e.g. $i_{1a}$) is significantly lower. Interestingly, the upper bound of $\sigma_{carlo,j}$ and $\sigma_{linear,j}$ for all parameters for $\sigma_{data} = 1.3$ is similar, close to 3.5 giving the maximum parameter estimation error to the experimental error ratio equal to $\ln(3.5)/\ln(1.3) \approx 4.8$ assuring reasonable identifiability of all parameters based on the simple on-off stimulation protocol.

The performed linear as well Monte Carlo indentifiability analyses are local in that sense that they start from the nominal parameter values. One can expect that for substantially different parameters the reduced model can be found non-identifiable.

## Characteristic behaviors

In this section we discuss the characteristic dynamical behaviors that can be exhibited by the reduced model (depending on the assumed parameters) in the response to the tonic TNF

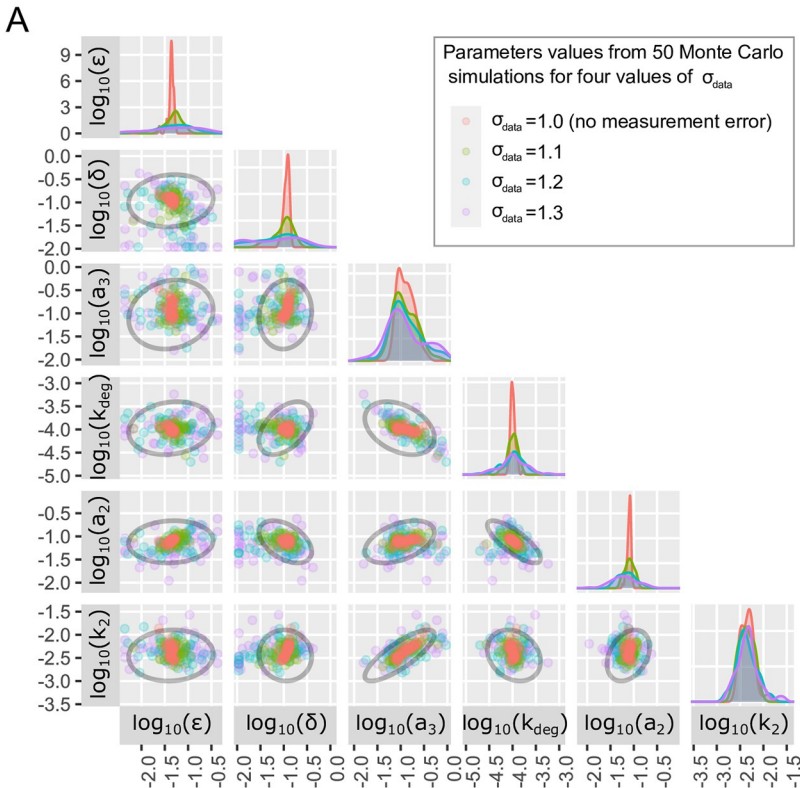

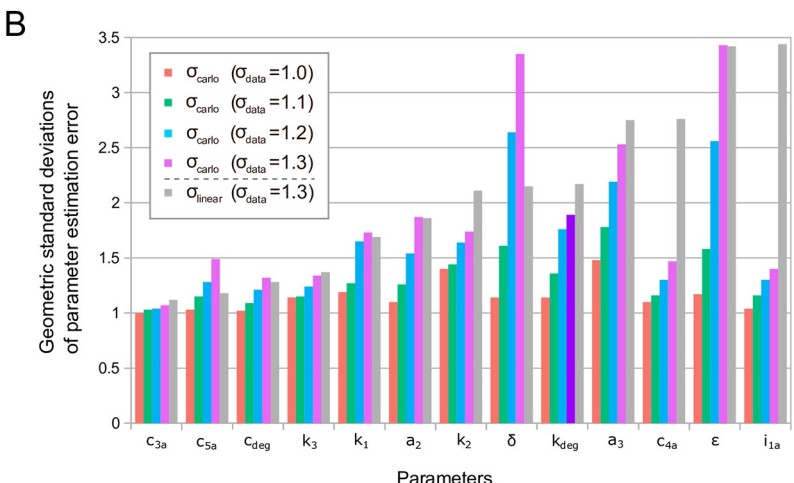

**Fig 7. Practical identifiability of the reduced model based on Monte Carlo simulations.** (A) A comparison of 75% confidence ellipses (shown in black) computed for $\sigma_{data}$ = 1.3 in the linear sensitivity matrix based analysis with results from 50 Monte Carlo simulations for four values of $\sigma_{data}$ (1.0, 1.1, 1.2, 1.3). Shown are projections on 15 planes spanned by 6 parameters with the largest geometric standard deviations $\sigma_{carlo,j}$ (for $\sigma_{data}$ = 1.3). (B) The geometric standard deviation of parameter estimation error $\sigma_{carlo,j}$ obtained in Monte Carlo simulation for four values of geometric standard deviations of measurements, $\sigma_{data}$ (1.0, 1.1, 1.2, 1.3), is compared with the geometric standard deviation of parameter estimation $\sigma_{linear,j}$ obtained from the linear analysis for $\sigma_{data}$ = 1.3.

stimulation. In Fig 8 prior to the tonic TNF stimulation at $t = 1$h, cells remain in the '$T_R = 0$' steady state in which ($IKKn$, $IKKa$, $NF\kappa B_n$, $A20$, $I\kappa B\alpha$, $I\kappa B\alpha_t$) = (1, 0, 0, 0, 0, 0).

For the nominal parameter values, i.e., fitted to the original model (see Table 1) the system exhibits damped oscillations and eventually converges to the '$T_R = 1$' steady state in which all six variables are greater than zero (see Fig 8A). The dynamics is characterized by a sharp increase of $NF\kappa B_n$ up to 1 (its maximal value). In the short $NF\kappa B_n$ increasing phase, $I\kappa B\alpha$ remains close to zero, and then starts increasing after $NF\kappa B_n$ reaches 1, see Fig 8A, right sub-panel.

The observed (damped) oscillations are a consequence of two negative feedbacks mediated by I$\kappa$B$\alpha$ (directly inhibiting NF-$\kappa$B) and A20 (inhibiting IKKa that targets I$\kappa$B$\alpha$ for degradation). The I$\kappa$B$\alpha$ feedback is associated with a time delay due to an intermediate step involving I$\kappa$B$\alpha$ transcript, and the process of I$\kappa$B$\alpha$ translocation. Increasing the strength of I$\kappa$B$\alpha$ mediated feedback (by decreasing the I$\kappa$B$\alpha$ degradation coefficients $a_2$ and $c_{5a}$) and the associated time delay (by decreasing the I$\kappa$B$\alpha$ nuclear import coefficient $i_{1a}$), we observe the appearance of limit cycle oscillations, Fig 8B. As expected, the amplitude of sustained I$\kappa$B$\alpha$ oscillations is much higher (about 3 fold with respect to the nominal parameters, Fig 8B, right sub-panel), and the period of oscillations is longer (the second $NF\kappa B_n$ peak is at about 5 not 3 hours, Fig 8B, left sub-panel).

A further decrease of parameter $a_2$ causes the growth of the oscillations amplitude, and changes their character, so they resemble spiky oscillations, Fig 8C. The right subpanel in Fig 8 shows the limit cycle, which consists of one 'slow' segment (in which $NF\kappa B_n \approx 0$) and one 'fast' segment or spike (away from $NF\kappa B_n \approx 0$).

Next, we investigate effects of A20 on the NF-$\kappa$B dynamics. We may observe that after removing A20 negative feedback by setting $k_2 = 0$ the reduced model does not exhibit any oscillations, but after slight overshooting reach '$T_R = 1$' the steady state characterized by a high level of nuclear $NF\kappa B_n$, Fig 9A. This is due to the fact that A20 mediated negative feedback is necessary for functioning of the primary negative feedback of I$\kappa$B$\alpha$, as only inhibition of IKKa allows for I$\kappa$B$\alpha$ accumulation. Finally, we consider consequences of a dramatic increase of both negative feedbacks resulting from the increase of parameters $k_2$ and $i_{1a}$. For the modified parameters we observe a system behavior close to the perfect adaptation [37]. An instant rise of stimulus (from $T_R = 0$ to $T_R = 1$) results in a spike of $NF\kappa B_n$ followed by a small amplitude damped oscillations converging to the steady state in which $NF\kappa B_n$ is close to zero as in the '$T_R = 0$' steady state, Fig 9B. The perfect adaptation means that the system responds to the temporal change of the stimuli rather than to the actual stimuli amplitude [38].

Clearly, $NF\kappa B_n$ dynamics can be dramatically changed by manipulating model parameters. The results presented above indicate that system (1–6) can produce characteristic dynamical responses to a TNF signal: a pulse, damped oscillations, or periodic oscillations (small-amplitude or spike-type). The system is able to respond in a non-adaptive or adaptive way, depending on the strength of negative feedbacks.

## Methods

### Normalization of *in silico* generated data

The applied normalization of the model generated data is based on assumption that experimental measurements are log-normally distributed, i.e., errors are proportional to the measured value of a given variable, provided that this value is above some threshold [36]. Hence, we normalize each simulated time series $x_i$ in two steps

1. replacing each $x_i$ by $x_i' = x_i + \rho \times \max(x_i)$, with $\rho = 0.03$.

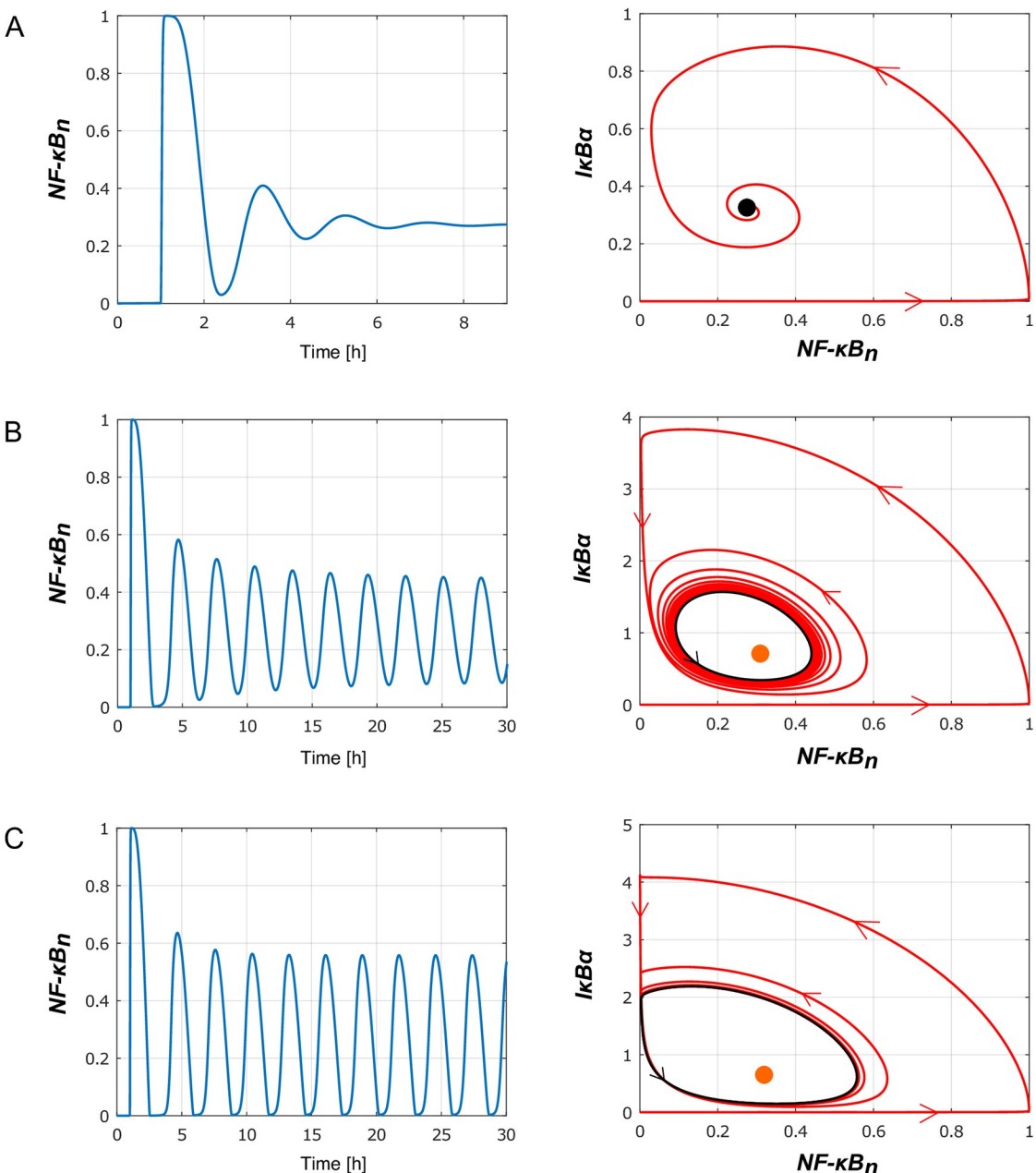

**Fig 8. Three types of the oscillatory responses of the reduced model to tonic TFN stimulation: Left subpanels show nuclear**
**$NF\kappa B_n(t)$ in response to tonic TNF stimulation started at $t$ = 1 hour, whereas the right subpanels show trajectories projected on**
**the ($NF\kappa B_n$, $I\kappa B\alpha$) plane.** (A) Damped oscillations observed for the nominal (fitted to the original model) parameters values (see
Table 1), in particular $a_2 = 0.0762$, $c_{5a} = 0.000058$, $i_{1a} = 0.000595$ (modified in panels B and C). The black dot on right subpanel
shows the '$T_R = 1$' stable steady state. (B) Limit cycle oscillations of nuclear $NF\kappa B_n(t)$ observed in simulations performed for $a_2 =$
$0.02$, $c_{5a} = 0.00001$, $i_{1a} = 0.0001$ and other parameters unchanged. The orange dot on the right subpanel shows the '$T_R = 1$' unstable
steady state surrounded by a stable limit cycle. A numerically computed orbit (shown in red) approaches the stable limit
cycle. (C) Periodic relaxation-like oscillations observed in simulations performed for $a_2 = 0.01$, $c_{5a} = 0.00001$, $i_{1a} = 0.0001$ and other
parameters unchanged. Again, the orange dot on the right subpanel shows the '$T_R = 1$' unstable steady state surrounded by a stable
limit cycle (black line).

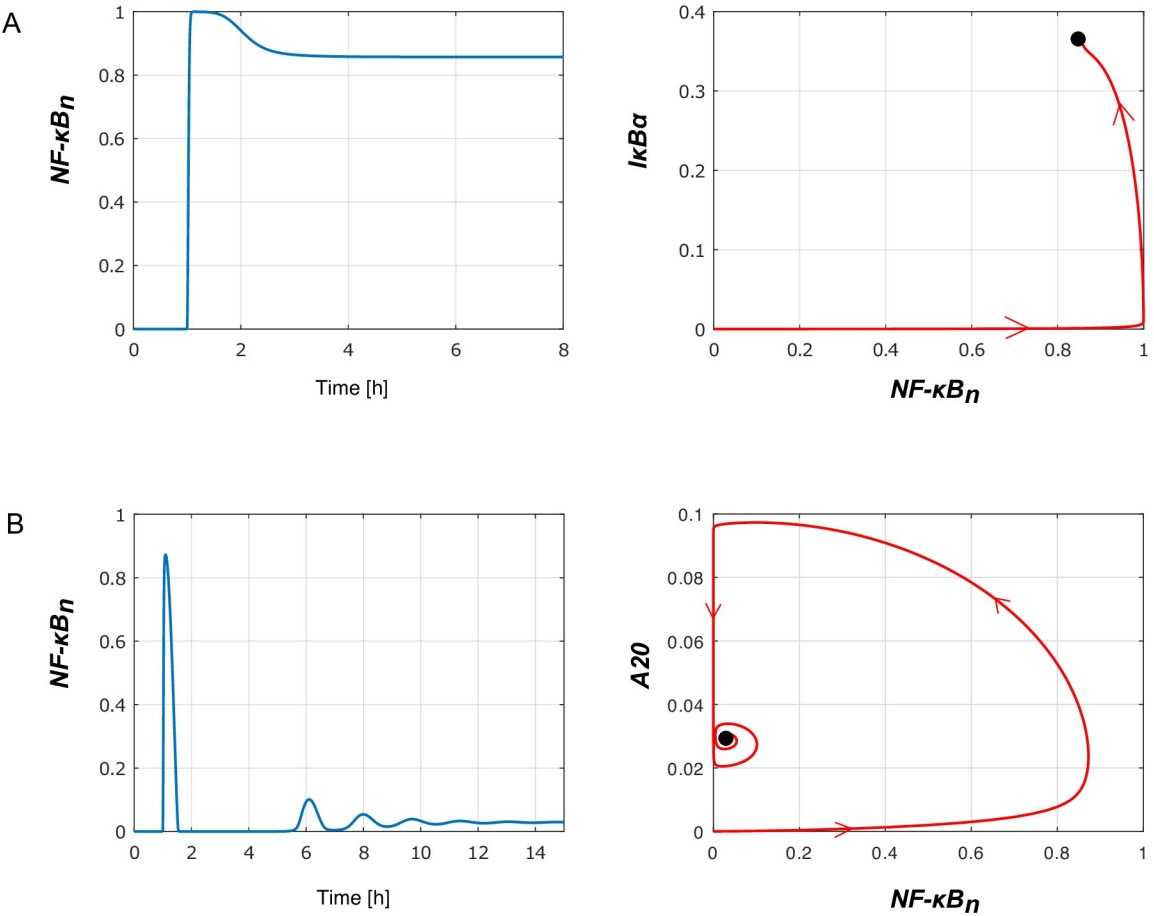

**Fig 9. Influence of A20 feedback on the reduced model responses to tonic TNF stimulation.** Left subpanels show nuclear $NF\kappa B_n(t)$ in response to tonic TNF stimulation started at $t = 1$ hour, whereas the right subpanels show trajectories projected on the ($NF\kappa B_n$, $I\kappa B\alpha$) or ($NF\kappa B_n$, A20) planes. (A) Numerically computed solution for $k_2 = 0$ (and other parameters unchanged, as in Table 1) implying absence of A20 mediated feedback (as in A20 KO cells). (B) Nearly-perfect adaptation observed in the model variant with significantly stronger negative feedbacks mediated by A20 and $I\kappa B\alpha$. Numerical simulations are performed for $k_2 = 3.57$ and $i_{1a} = 0.01$ (changed from the nominal values $k_2 = 0.0357$ and $i_{1a} = 0.00595$) and other parameters unchanged.

2. replacing each $x_i'$ by $y_i = x_i' / (\Pi_{i=1}^n x_i')^{\frac{1}{n}}$.

The first normalization step is needed to avoid values smaller than $0.03 \times \max(x_i)$ that typically represent nothing, but measurement noise. The second normalization step reflects the observation that most of the currently available data is in arbitrary units, and thus must be normalized. Normalization by geometric (not arithmetic) mean of time series, reflects the assumption that multiplicative changes of all variables are more important than additive change. An immediate consequence of normalization is that a time series of $n$ measurements of a given variable gives $n - 1$ independent numbers.

## Model fitting

The original and reduced models are both implemented in BioNetGen (cvode solver) and MATLAB. The simulated model outputs are normalized as described above.

In order to fit the reduced model to the original model, we use PyBioNetFit [34], where the corresponding parameter estimation problem in the least-squares sense for the log-

transformed systems is formulated [39, 40]. More precisely, the parameter estimation proce-
dure finds a parameter set minimizing the following objective function

$$J(\tilde{\theta}) = \sum_{i=1}^{N} (\ln y_i^R - \ln y_i^O)^2, \qquad (7)$$

where $y_i^R$ denotes the reduced model outputs, $y_i^O$ denotes the original model outputs, and *N* is
the number of all measurements in the considered protocol. This objective function quantifies
the deviation of the (log-transformed) reduced model trajectories from the trajectories of the
(log-transformed) original model.

The Average Multiplicative Distance (AMD) between the original and the reduced model
(either fitted or not fitted) is defined as

$$\text{AMD} = \exp\left( \sqrt{\frac{1}{N} \sum_{i=1}^{N} (\ln y_i^R - \ln y_i^O)^2} \right) \qquad (8)$$

with *N* denoting the number of measurements. We denote by $\text{AMD}_{WT}$ and $\text{AMD}_{A20KO}$ the
average multiplicative distance for WT and A20KO cells, respectively.

## Linear identifiability analysis

For the linear identifiability analysis [41–44] we use a sensitivity matrix defined as $S = s_{ij}$,
where $s_{ij} = \frac{\partial \ln(y_i)}{\partial \ln(\theta_j)} = \frac{\partial y_i}{\partial \theta_j} \frac{\theta_j}{y_i}$ measures a multiplicative change of data value $y_i$ with respect to a mul-
tiplicative change of the parameter $\theta_j$ ($j = 1, 2, \ldots, p$), where *p* is the number of parameters,
equal to 13 for the reduced model and 23 for the original model. The matrix *S* is calculated
based on the considered stimulation protocols with the corresponding series of measurements
of the five reduced model variables (as detailed in S1 Table and Table 2). The resulting sensitiv-
ity matrix consists of 13 (for the reduced model) or 23 (for the original model) sensitivity vec-
tors, $S_j$, each of length equal to the number of data points in a given experiment or series of
experiments. The elements $s_{ij}$ of the sensitivity matrix are calculated numerically using the
finite differences method [45], with 1% increase/decrease of parameter values. In our analysis,
parameters were individually perturbed by a 1% increase/decrease of the original values.

The singular values are calculated by singular value decomposition (SVD) [3, 46, 47]. The
perpendicular parts $S_j^{\perp}$ of sensitivity vectors $S_j$ are calculated as $S_j^{\perp} = S_j - P_{*,j} S_j$. The matrix
$P_{*,j} = S_{*,j} (S_{*,j}^T S_{*,j})^{-1} S_{*,j}^T$ is the so-called projection matrix, where $S_{*,j}$ denote a matrix formed by
removing the *j*-th sensitivity vector from the full sensitivity matrix *S*.

To calculate ratio of parameter estimation error to the experimental error, we assume that
experimental data points errors are log-normally distributed, each with the same standard geo-
metric deviation $\sigma_{data}$. As a consequence one may expect that the corresponding parameter
estimates are also log-normally distributed with standard geometric deviation $\sigma_{linear,j}$, where
$j = 1, \ldots, p$ is the parameter index. By definition, $R_j = \ln(\sigma_{linear,j})/\ln(\sigma_{data})$. Based on the equation
(3.20) in [48], the random variable $\xi_{par} = [\xi_{par,1}, \ldots, \xi_{par,p}]$ describing the (linear) parameter
estimation error is given by the following formula:

$$\xi_{par} \approx (S^T S)^{-1} S^T \xi_{obs}, \qquad (9)$$

where $\xi_{obs} = [\xi_{obs,1}, \ldots, \xi_{obs,m}]$ is a random variable describing the experimental error. Hence,
the ratio $R_j$ of the parameter estimation error to the experimental error can be calculated as the
$l_2$-norm of the *j*-th row vector of the *S* pseudo-inverse matrix $A := (S^T S)^{-1} S^T$. We used a math-
ematically equivalent, but more numerically stable, calculation of $R_j$ based on *QR*-

decomposition of $S$ (see [39]). In this approach the ratio $R_j$ is given by the $l_2$-norm of the $i$-th row vector of the $R_a^{-1}$ matrix, where $R_a$ denotes the matrix composed of the first $p$ rows of an upper triangular matrix $R$ from the $QR$-decomposition of the sensitivity matrix $S$.

### Monte Carlo based identifiability analysis

The geometric standard deviation of parameter estimation in Monte Carlo simulations is calculated as

$$\sigma_{carlo,j} = \exp[SD(\ln(\theta_{j,k}))], \tag{10}$$

where $SD$ denotes a standard deviation, and $\theta_{j,k}$ are $k = 50$ numerical estimates of parameter $\theta_j$, $j = 1, \ldots, p = 13$, based on model refitting to the $k$ perturbed trajectories.

## Discussion

Mechanistic models of signaling pathways, including the NF-$\kappa$B pathway, are typically non-identifiable, meaning that their parameters may not be determined based on existing data [49]. In simple words, by fitting a non-identifiable model to experimental data the researcher demonstrates the model ability to reproduce biological observations, but get no or little insight into parameters describing the regulatory process. When a model is structurally non-identifiable, an arbitrary large change of a given parameter can be compensated by an appropriate change of some other parameter (or parameters) rendering this parameter pair (set) non-identifiable, whereas practical non-identifiability arises when parameters may not be determined based on available data with adequate precision. It is thus desirable to reduce mechanistic models in such a way that they still capture dynamics of the observed variables, but reach the stage at which all their parameters become identifiable. One should notice, however, a frequent drawback of reduced identifiable models, i.e., the presence of "composite" parameters that do not represent biochemical constants.

Resolving non-identifiability involves a number of steps, some straightforward, whereas some requiring understanding of model dynamics, which can hardly be pursued according to a simple rule-based algorithm. The most obvious step is model non-dimensionalization. Indeed, when experimental data give only the relative changes of model variables, the dimensional model may not be identifiable, because it "predicts" absolute values that are not measured. The second step can involve the removal of equations for fast variables by assigning the steady-state values to corresponding variables. This step requires at least the approximate knowledge of time scales of modeled processes. The third step may require the removal of some intermediates that are not experimentally measured. This step is more challenging, because these intermediates contribute to time delays between observed variables. Importantly, a series of intermediates is associated with a time delay that follows an Erlang-type distribution, and thus may not be reproduced by a single process associated with an exponential distribution of time delay.

In this study we focused on the deterministic model of the NF-$\kappa$B pathway, which contains 15 equations, and in which temporal profiles of nuclear NF-$\kappa$B are regulated by two interlinked negative feedback loops mediated by I$\kappa$B$\alpha$ and A20. Based, essentially, on the three above-outlined types of reductions, we reduced the considered model to 6 equations parameterized by 13 parameters. We have demonstrated that the reduced model can satisfactorily reproduce the behavior of the original model in response to the tonic stimulation and five pulsatile experimental protocols (studied before). Moreover, the reduced model can reproduce (after refitting) the original model behavior even when parameters of the original model are randomly changed up to threefold from their original values (which may be needed to

reproduce behaviors of different cell types). Next, we demonstrated that the reduced model is structurally and practically identifiable based on experimental protocols and measurements that have been previously applied to the analysis of NF-κB pathway. This implies that the reduced model conserves dynamical properties of the original model, but in contrast to the original model, its parameters can be determined. In particular, we showed, theoretically, that a relatively simple on–off protocol in which 2 hour-long TNF stimulation is followed by a 10 hour-long TNF wash-out phase, and in which the main five models variables are measured, suffices to identify model parameters. The on–off protocol of properly chosen lengths of the on and the off phases allows to determine both the forward and backward rates, and thus can be considered as a first choice protocol to investigate dynamics of regulatory systems. We verified practical parameter identifiability for this protocol with the help of Monte Carlo simulations in which we refitted the model parameters to perturbed observables. The obtained parameter estimation errors follow unimodal distributions with geometric standard deviations comparable to those predicted by the linear analysis based on the sensitivity matrix. This indicates that the reduced model can be fitted to noisy data.

Finally, we demonstrated that the reduced model may exhibit (depending on the assumed parameters controlling negative feedbacks) different types of responses characteristic to regulatory motives controlled by negative feedback loops [50]: nearly-perfect adaptation, damped or sustained oscillations. These responses may allow to reproduce and explain behaviors of different cells types.

In the last years various techniques to resolve "easier" structural and "harder" practical identifiability have been developed (reviewed in [1]). Even if some of these techniques have been tested on the Lipniacki et al. 2004 model [5]: see [2, 9, 10] or its later variant (from [51]) [11], to the best of our knowledge, none of these models have been demonstrated to simultaneously exhibit practical identifiability based on selected experiment (or experiments) and to reproduce experimentally verified trajectories of the original models (reviewed in Ref. [17]). The first reduced model (based on the pioneering Hoffmann et al. 2002 model), containing three variables (free nuclear NF-κB, IκBα mRNA and free cytoplasmic IκBα), was proposed by Krishna et al. [6]. Since the model is non-dimensional and contains only 5 parameters, it is likely identifiable based on existing data for WT cells, but since it has no A20-mediated feedback loop, it may not distinguish between WT and A20 KO cells. Because the Krishna et al. [6] model accounts only for IKK-driven IκBα degradation (parameter $C$ governing IκBα degradation is assumed to be proportional to IKK considered as a stimulus), in the absence of stimulation, IκBα may remain on an arbitrarily high level. As a consequence, the model may not reproduce responses to TNF pulses observed by Ashall et al. [21], just because at the second TNF pulse the level of IκBα is very high. We found, however, that when parameter $C$ is assumed constant, while parameter $A$, proportional to the source term for nuclear NF-κB, is considered as an input, the model possesses a natural (0,0,0) steady state in the absence of stimulation. Under such interpretation the model may be fitted to the Lipniacki et al. 2002 model [5] and satisfactorily reproduce NF-κB and IκBα mRNA profiles, however, the free cytoplasmic IκBα profile is different, S8 Fig. Consequently, AMD between these two models exceeds 2, implying a relatively poor fit. The Krishna et al. 2006 model [6] was later supplemented by equations for active and inactive IKK [7], but still the resulting 5-variable model neglects the A20-negative feedback loop (A20 is assumed to be constant). Moreover, the model is presented in a dimensional form, and thus may not be identifiable based on data without absolute mRNA and protein levels measurements. Another 5-variable model (without A20) was constructed by Zambrano et al. [8], but neither its identifiability nor responses to pulsatile TNF stimulation (the authors assume exponential decrease of the active IKK fraction) were analyzed.

In summary, we demonstrated the possibility to reduce the NF-*κ*B pathway model to a model that is identifiable, but still capable of reproducing rich dynamics of the original model in response to tonic as well as pulsatile TNF stimulation. Our approach can hardly be automated, as it uses intuitive knowledge about the dynamics of the regulatory system and time scales associated with the involved biochemical processes. Nevertheless, we hope that our study opens a new perspective in modeler-assisted model reduction, and will help to represent other regulatory pathways in terms of identifiable models. Such representation brings the system biology methods closer to those characteristic for physics, where the models, even if approximate, have well-defined parameters. The proposed reduced NF-*κ*B pathway model can serve as a building block for more comprehensive models of the innate immune response and cancer, where NF-*κ*B plays a key regulatory role.

## Supporting information

**S1 Fig. Comparison between the Lipniacki et al. 2004 model and the Hoffmann et al. 2002 model in two variants: With and without I*κ*B*β* and I*κ*B*ϵ* isoforms.** The details for Hoffmann et al. 2002 model simulations are provided in S1 Text.
(PDF)

**S2 Fig. Comparison between Lipniacki et al. 2004 model, Ashall et al. 2009 model, and Murakawa et al. 2015 model in WT cells.** The details for Ashall et al. 2009 model and Murakawa et al. 2015 model simulations are provided in S1 Text.
(PDF)

**S3 Fig. Comparison between Lipniacki et al. 2004 model, Ashall et al. 2009 model, and Murakawa et al. 2015 model in KO A20 cells.** The details for Ashall et al. 2009 model and Murakawa et al. 2015 model simulations are provided in S1 Text.
(PDF)

**S4 Fig. Dynamics of the original, reduced and reduced fitted models in combination experiment in A20 KO cells defined in S1 Table.** Red dots indicate time points from which the nuclear NF-*κ*B and total I*κ*B*α* protein *in silico* measurements are used for fitting the reduced model to the original one. The time points for the remaining variables are given in S1 Table.
(PDF)

**S5 Fig. Simulations of the original model for A20 KO cells for five different sets of parameters and the corresponding reduced model with refitted parameter values (see S3 and S4 Tables).** Simulations are performed for the combination experiment defined in S1 Table. For each parameter set the corresponding $AMD^*_{A20KO}$ is computed based on trajectories of the 5 main model variables.
(PDF)

**S6 Fig. Dynamics of the reduced fitted model for the on-off protocol in WT cells defined in Table 2.**
(PDF)

**S7 Fig. Practical identifiability of the reduced model based on Monte Carlo simulations.** A comparison of 75% confidence ellipses (shown in black) computed for $\sigma_{data}$ = 1.3 in the linear sensitivity matrix based analysis with results from 50 Monte Carlo simulations for four values of $\sigma_{data}$ (1.0, 1.1, 1.2, 1.3). Shown are projections on 21 planes spanned by 7 parameters with the smallest geometric standard deviations $\sigma_{carlo,j}$ (for $\sigma_{data}$ = 1.3).
(PDF)

**S8 Fig. Fit of the 'reinterpretted' Krishna et al. 2006 model to Lipniacki et al 2004 model.**
In contrast to the original Krishna et al. assumption, it is assumed that the parameter $C$ (governing I$\kappa$B$\alpha$ degradation) remains constant, while the parameter $A$ (multiplicating the source term for NF-$\kappa$B) attains its nominal value, when TNF is on, and zero value, when TNF is off. The trajectories are plotted in log scale; dots denote fitting time points, and the trajectories are plotted as saw lines between these points. The discrepancy between free cytoplasmic I$\kappa$B$\alpha$ trajectories causes that average multiplicative distance between these two models equals 2.06.
(PDF)

**S1 Table. Summary of the considered stimulation protocols of the NF-*κ*B system in WT and A20 KO cells used for *in silico* experiments.** Each table block describes one TNF stimulation protocol for which the measured variables are: IKKa, nuclear NF-$\kappa$B, A20, total I$\kappa$B$\alpha$ protein denoted by I$\kappa$B$\alpha$* (I$\kappa$B$\alpha$* = I$\kappa$B$\alpha$ + 1 − NF$\kappa$B), and I$\kappa$B$\alpha$ mRNA. For each variable the measurement times used for the sensitivity-based identifiability analysis and reduced model fitting are given.
(PDF)

**S2 Table. List of protocol sets, the corresponding numbers of measurement time points (N), and sensitivity vectors dimensions (dim).** Protocol sets are joint for WT and A20 KO. The combination experiment consists of 6 protocols: continuous and all pulsatile ones. The scaling factor is $\sqrt{dim}$.
(PDF)

**S3 Table. Parameter sets of the original model.** Five parameter sets of the original model randomly selected by varying the original parameters at most three-fold from their original values.
(PDF)

**S4 Table. Fitted parameter sets of the reduced model obtained from fitting to the original model with varied parameter sets.**
(PDF)

**S1 Text. Details of the NF-*κ*B models simulations.**
(PDF)

**S2 Text. Model reduction, scaling and non-dimensionalization.**
(PDF)

**S1 Code. Numerical codes for computational NF-*κ*B models: Reduced (developed in this study), Hoffmann et al. 2002 [16], Lipniacki et al. 2004 [5], Krishna et al. 2006 [6], Ashall et al. 2009 [21] and Murakawa et al. 2015 [32].**
(ZIP)

## Acknowledgments

Numerical simulations were performed using PLGrid Infrastructure. We thank Dr Marek Kochańczyk for revision of computational codes.

## Author Contributions

**Conceptualization:** Joanna Jaruszewicz-Błońska, Ilona Kosiuk, Wiktor Prus, Tomasz Lipniacki.

**Formal analysis:** Joanna Jaruszewicz-Błońska, Ilona Kosiuk.

**Methodology:** Joanna Jaruszewicz-Błońska, Ilona Kosiuk.

**Software:** Joanna Jaruszewicz-Błońska.

**Visualization:** Joanna Jaruszewicz-Błońska, Ilona Kosiuk.

**Writing – original draft:** Joanna Jaruszewicz-Błońska, Ilona Kosiuk, Tomasz Lipniacki.

**Writing – review & editing:** Joanna Jaruszewicz-Błońska, Ilona Kosiuk, Wiktor Prus, Tomasz Lipniacki.

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
