## [Decision Letter · Decision Letter 0]

3 Apr 2023

PONE-D-23-07010A plausible identifiable model of the canonical NF-κB signaling pathwayPLOS ONE

Dear Dr. Lipniacki,

Thank you for submitting your manuscript to PLOS ONE. After careful consideration, we feel that it has merit but does not fully meet PLOS ONE’s publication criteria as it currently stands. Therefore, we invite you to submit a revised version of the manuscript that addresses the points raised during the review process.

We look forward to receiving your revised manuscript.

Kind regards,

Jordi Villà-Freixa

Academic Editor

PLOS ONE

Journal Requirements:

   "This research was supported by National Science Centre (Poland) grant 2018/29/B/NZ2/00668 and Norwegian Financial Mechanism GRIEG-1 grant (operated March 7, 2023 25/31 by the National Science Centre, Poland) 2019/34/H/NZ6/00699. Numerical simulations were performed using PLGrid Infrastructure. During the initial stage of the project IK was supported by the European Union’s Horizon 2020 research and innovation program under the Marie Sklodowska-Curie Grant Agreement No 661650. IK thanks Vienna University of Technology for support and hospitality."

4. Please remove your figures from within your manuscript file, leaving only the individual TIFF/EPS image files, uploaded separately. These will be automatically included in the reviewers’ PDF.

Additional Editor Comments:

There are some minor to major improvements suggested by the referees, but it is critical that you speciffically address the concerns raised by referee 3, in particular: "The main strength put forward by the authors that their study constructs a reduced, identifiable model and ‘to the best of their knowledge the first accounting for two negative feedback loops (mediated by IkBalpha and A20)' is not correct. Mathematical models for the NF-kB system have been recently reviewed in a paper of Haga and Okada, Biochemical Journal 2022, 279. 161 who report on several NF-kB models including the A20 feedback beside the IkBalpha feedback. Highly relevant for the approach introduced here seems the paper of Mothes et al. Frontiers Physiology 2020 11, 896 that determined the impact of different A20 implementations on the dynamics of NF-kB in models of a comparable size as the one developed here.

A comparison with the other minimal models should include these models as well. "

Reviewers' comments:

Reviewer's Responses to Questions

**Comments to the Author**

1. Is the manuscript technically sound, and do the data support the conclusions?

Reviewer #1: Yes

Reviewer #2: Yes

Reviewer #3: Yes

2. Has the statistical analysis been performed appropriately and rigorously? 

Reviewer #1: Yes

Reviewer #2: Yes

Reviewer #3: N/A

3. Have the authors made all data underlying the findings in their manuscript fully available?

Reviewer #1: No

Reviewer #2: Yes

Reviewer #3: Yes

4. Is the manuscript presented in an intelligible fashion and written in standard English?

Reviewer #1: Yes

Reviewer #2: Yes

Reviewer #3: Yes

5. Review Comments to the Author

Reviewer #1: In this paper the authors reduce an existing model of the canonical NF-κB pathway from 15 to 6 equations

The reduced model retains two negative feedback loops mediated by IκBα and A20.

A sensitivity-based linear analysis and a Monte Carlo-based analysis demonstrates that the resulting model is both

structurally and practically identifiable given measurements of 5 model variables from a simple TNF stimulation protocol.

The reduced model is capable of reproducing different types of responses that are characteristic of regulatory motifs controlled by negative feedback loops including nearly-perfect adaptation as well as damped and sustained oscillations.

The paper is well considered, well written and the authors offer excellent evidence and analyses that they have reduced the original NF-κB pathway model to a model that is identifiable, but still capable of reproducing the rich dynamics of the original model in response to tonic as well as pulsatile TNF stimulation. The approach and new model makes a significant contribution to the field of Systems Biology.

Reviewer #2: In this interesting work Jaruszewicz-Blonska et al. propose a new mathematical model of the canonical NFkB system by drastically reducing the dimensions of a previous existing model. After this, they perform a number of numerical simulations to assess the 'identifiability' of the model. In particular they show that different combinations of simulated data from the original model lead to a better 'identifiability' when fitted with the reduced model than with the original one, while being able to reproduce the dynamics of the original model with reasonable accuracy. Very importantly, they are able to identify potential “experimental protocols” that would produce data that would make the reduced model identifiable. The reduced model is also identifiable for in-silico data blurred by noise. Finally, it is shown that the reduced model displays the qualitative dynamics observed in different experimental contexts, including damped and persistent oscillations.

Although model reduction should in principle contribute to a better identifiability of the parameters, to my knowledge this is the first time in which this has been systematically explored for NFkB signaling models, and the fact that the proposed model is more identifiable for this in-silico data carries important insights to be considered when opting between simple or complicated models of a signaling system. I also found of particular interest the fact that they show how specific “experimental” protocols might lead to better identifiability. The paper is well written and in my opinion technically very sound.

My main (yet minor) concern is related with the way authors use the concept of “identifiability”. In my opinion, identifiability refers always to certain kind of data to which the model shall be fit, and what we can say for their reduced model is that the model is identifiable when trying to fit in-silico data produced by the original model. This is something that, I underline again, is of theoretical interest by itself and potentially useful for applications, e.g. to fitting to real experimental data, something that probably is beyond the scope of the manuscript. But I think that this should be stated more explicitly.

This is an issue that is somehow indicated by the previous reviewers of the manuscript (I thank the authors for including these reports): Reviewer 1, raises in point 3 how “it is unclear that the reduced model is capable of recapitulating experimental data”. The authors reply saying that their reduced model can qualitatively reproduce the original model, which itself was “based on experimental data”. However, it is unclear to me if the original model (and hence the reduced one) was able to qualitatively or quantitatively (with a small fitting error) reproduce experimental data. The figure shown indicates the former, not the latter. It could well be that when trying to fit experimental data to the reduced model, the discrepancy is high and some parameters might not be identifiable. Hence I’d suggest that the authors discuss this in more detail and to specify more clearly (maybe also in the title) that the reduced model is identifiable with respect to in-silico data generated from the higher dimensional original model. I think that this way their interesting contribution will become more clear.

A second related minor concern is that the reduced model relies on approximations on the original model that in some cases are based on the existence of “fast” processes. However I speculate that the ability to identify the parameters and closely reproduce the original model with high accuracy might be limited to ranges of the original parameter set for which these reactions can still be considered “fast”. This might not need to be tested with additional simulations, but I’d like to see this discussed in the manuscript.

Overall I think that this is an interesting manuscript that might require small corrections to put on a clearer light the contribution to the NFkB modeling field made by the authors.

Reviewer #3: The manuscript entitled “A plausible identifiable model of the canonical NFκB signaling pathway” by Jaruszewicz-Błońska et al. already went through a major revision. While I was not part of the review process I agree with the three reviewers that the work described in the manuscript is ‘probably technically sound’, but there 'substantial concerns about the approach and ultimate interest'. While I find the questions of the reviewers highly relevant these are not fully answered by the authors.

The main strength put forward by the authors that their study constructs a reduced, identifiable model and ‘to the best of their knowledge the first accounting for two negative feedback loops (mediated by IkBalpha and A20)' is not correct. Mathematical models for the NF-kB system have been recently reviewed in a paper of Haga and Okada, Biochemical Journal 2022, 279. 161 who report on several NF-kB models including the A20 feedback beside the IkBalpha feedback. Highly relevant for the approach introduced here seems the paper of Mothes et al. Frontiers Physiology 2020 11, 896 that determined the impact of different A20 implementations on the dynamics of NF-kB in models of a comparable size as the one developed here.

A comparison with the other minimal models should include these models as well.

Also the question (reviewers 1 major points) whether the reduced model can recapitulate experimental data (e.g. from literature) is not convincingly addressed.

In that context I find the argumentation of the authors to reviewer 2 with respect to ‘non-identifiable models’ as ‘typically containing wrong parameters’, arguing ‘thus it is not clear which of the predictions are true and which not’ contradictory to the approach to fit the model to the simulated dynamics of exactly such ‘non-identifiable models’- this seems to be a contradictory approach that needs clarification.

6. PLOS authors have the option to publish the peer review history of their article (what does this mean?). If published, this will include your full peer review and any attached files.

Reviewer #1: No

Reviewer #2: No

Reviewer #3: No

---

## [Author Response · Author response to Decision Letter 0]

12 May 2023

The response to reviewer and editor comments is provided as the attached document: Jaruszewicz_et_al__Resubmission__Response_to_Reviews.pdf

---

## [Editor Report · Decision Letter 1]

16 May 2023

A plausible identifiable model of the canonical NF-κB signaling pathway

PONE-D-23-07010R1

Dear Dr. Lipniacki,

We’re pleased to inform you that your manuscript has been judged scientifically suitable for publication and will be formally accepted for publication once it meets all outstanding technical requirements.

Kind regards,

Jordi Villà-Freixa

Academic Editor

PLOS ONE
---

## [Editor Report · Acceptance letter]

23 May 2023

PONE-D-23-07010R1 

A plausible identifiable model of the canonical NF-κB signaling pathway 

Dear Dr. Lipniacki:

I'm pleased to inform you that your manuscript has been deemed suitable for publication in PLOS ONE. Congratulations! Your manuscript is now with our production department. 

Kind regards, 

on behalf of

Dr. Jordi Villà-Freixa 

Academic Editor

PLOS ONE